# Considerable Variation in Intake of Live Food Microbes in Dutch Adults

**DOI:** 10.3390/nu17071248

**Published:** 2025-04-03

**Authors:** Berber J. Vlieg-Boerstra, Jeanne H. M. de Vries, Anastriyani Yulviatun, Marcel H. Zwietering

**Affiliations:** 1Department of Pediatrics, OLVG Hospital, 1090 HM Amsterdam, The Netherlands; b.vlieg-boerstra@olvg.nl; 2Rijnstate Allergy Centre, Rijnstate Hospital, 681 5AD Arnhem, The Netherlands; 3Vlieg Dieticians, Nutrition and Allergy, 6815 AD Arnhem, The Netherlands; 4Division of Human Nutrition and Health, Wageningen University & Research, P.O. Box 17, 6700 AA Wageningen, The Netherlands; jeanne.devries@wur.nl (J.H.M.d.V.); anastriyani@staff.uns.ac.id (A.Y.); 5Food Microbiology, Wageningen University & Research, P.O. Box 17, 6700 AA Wageningen, The Netherlands; 6Department of Food Science and Technology, Faculty of Agriculture, Sebelas Maret University, Surakarta 57126, Indonesia

**Keywords:** live food microorganisms, microbial exposure, lactic acid bacteria, diet, allergic disease

## Abstract

**Background:** Diet is an important source of microbial exposure, potentially protecting against allergic disease. However, changes in dietary habits may have altered the intake of live food microbes. **Aim:** We quantitatively assessed the intake of live food microbes in Dutch adults and compared these estimates with those obtained from duplicate portions. **Methods:** In 34 Dutch adults (20–70 years), we estimated the food-microbial content of their diet for three dominant groups: total contaminating bacteria (TCB), lactic acid bacteria (LAB), and yeasts/moulds (YM). A food-microbial load database was compiled with minimum, best, and maximum estimated levels of these food microbes (in colony forming units (CFU)/g food). To estimate microbial intake, the amounts of food consumed (in grams) based on three 24 h dietary recalls were multiplied by the corresponding microbial content/gram of food. For validation, one 24 h duplicate portion per person was analysed for microbial content by conventional plate counting. We applied a one-way ANOVA to assess interindividual variation in microbial exposure, a two-way ANOVA to assess intraindividual variation in microbial intake, the so-called MOM2 analysis and regression analysis to identify foods contributing most to the level and variation of microbial intake, and Bland–Altman plots to assess the agreement between microbial intake estimated from 24-HDR and microbial counts determined in duplicate food samples. **Results:** The estimated total microbial exposure varied considerably across individuals (*p* < 0.05), ranging from 5.7 to 11.6 log (5.4 × 10^5^–4.4 × 10^11^) CFU/day. The exposure to TCB ranged from 2.5 to 11.4 log (3.0 × 10^2^–2.5 × 10^11^) CFU/day, LAB from 3.4 to 11.5 log (2.3 × 10^3^–3.0 × 10^11^), and YM from 2.6 to 9.6 log (3.6 × 10^2^–4.3 × 10^9^) CFU/day. Also, intraindividual exposure differed significantly (*p* < 0.05). Estimates were confirmed by microbial analysis of 24 h duplicate foods, revealing total microbial levels of 6.4–11.0 log (2.8 × 10^6^–9.4 × 10^10^) CFU/day. On average, the best estimated total microbial exposure was 0.6 log CFU higher than values determined by plate counting of the duplicate foods. Foods responsible for the observed variation were identified. **Conclusions:** The intake of live food microbes among Dutch adults varied considerably, ranging from nearly a million to more than 100 billion cells per day. Further validation of the food-microbial database is required.

## 1. Introduction

The prevalence of allergic disease, such as atopic dermatitis, allergic asthma, allergic rhinitis, food allergy and eosinophilic esophagitis, has increased over recent decades [1]. Changes in the gut microbiome, an important modulator for the maturation of the immune system, may be a contributing factor [2,3]. A Western lifestyle is characterised by dysbiosis and decreased diversity of the (gut) microbiome [3,4]. A low diversity of the gut microbiome has been associated with the development of atopic disease [5,6,7]. It has been proposed that reduced exposure to environmental microbes contributes to microbial dysbiosis, inflammation, and an increased risk of allergic diseases [4]. In contrast, exposure to a diversity of microbes potentially protects against the development of allergic disease [8,9].

Diet is a major determinant of the gut microbiota, not only by providing nutrients and dietary fibre as substrates for microbial growth [10] but also by directly introducing live food microbes, which increases microbial exposure in the gut [11]. The potential benefits of live food microbes, particularly from fermented foods, have long been recognised. The anecdotal health benefits of Bulgarian yoghurt leading to longevity were already mentioned by Metchnikoff as early as in 1907 [12]. Today, the overall health benefits of live food microbes, particularly their anti-inflammatory effects and influence on microbiome diversity, are increasingly recognised [13,14,15,16,17,18,19,20].

In contrast to probiotics in capsules, the interaction between food microbes and immune cells, such as regulatory antigen-presenting cells and regulatory T cells, is not restricted to the small intestine but already begins in the mouth. Not all ingested food microbes are killed in the stomach’s acidic environment. Like probiotics, food microbes can be detected in the stool, indicating their partial survival through the digestive tract [11,14,17]. In adults, food microbes make up, on average, around 3% of the adult gut microbiome abundancy, while in infants, it is about 8% [11].

Given that diet is a modifiable factor, it is a promising target for interventions [15,21]. However, studies on the role of live microbes in allergic diseases have primarily focused on probiotic supplementation, with results often being inconsistent [22,23]. This may be caused by strain-specific effects of probiotics, whereas the diverse range of food microbes could potentially induce larger effects [22].

Several studies have demonstrated that the consumption of yoghurt, cheese, and freshly prepared homemade foods can have protective effects on the development of allergic diseases in early life by [24,25,26]. These protective relationships could at least partly be explained by the presence of food microbes, which could have a tolerogenic influence on the immune response. This supports the hypothesis that food microbes could be considered as important nutritional determinants in promoting a healthy diet [27] and reducing the risk of allergic disease.

The advent of next-generation sequencing has provided modern techniques for the characterization of the diversity in food microbial intake [11]. However, it is not only the types but also the quantity of food microbes, that determine the total food microbial intake and immune homeostasis. Currently, little is known regarding the overall consumption of food microbes and their variation [11,28].

The purpose of this study was to quantitatively assess the total consumption of food microbes in Dutch adults, and to examine the intra- and interindividual variation of intake of three dominant microbial groups. We also aimed at identifying the foods that contribute most to this intake and its variation, as well as to compare the estimated microbial intake to the microbial load measured in duplicate food portions.

## 2. Materials and Methods

### 2.1. Study Population and Consumption Data

The study population consisted of 34 Dutch adults (aged 20–70 years) living in Wageningen, a small city in the central Netherlands. Food consumption data were obtained through three telephone-based 24 h dietary recalls (24-HDRs) and one 24 h duplicate food sample (described below). These data were collected as part of the DUPLO study. This study was performed between 2011 and 2014 by the Division of Human Nutrition and Health of Wageningen University [29]. The DUPLO study was conducted according to the guidelines laid down in the Declaration of Helsinki and was approved by the medical ethical committee of Wageningen University (ID 34775, 11 July 2011). Written informed consent was obtained from all participants [29].

### 2.2. Compilation of a Food Microbial Content Table

Three major groups of food microbes were selected for microbiological analyses: total contaminating bacteria (TCB), lactic acid bacteria (LAB), and yeasts/moulds (YM). These groups of microbes are the main contributors to the total dietary microbial exposure. TCB are bacteria that can grow under aerobic conditions and are naturally present in food. They are often responsible for spoilage during storage and encompass many different types of food microbes. LAB are of specific interest as they are typically the dominant microbial group in many foods. LAB are intentionally added to foods, like yoghurt and sauerkraut, where they proliferate in large quantities. They also play an important role in spoilage of food, specifically in vacuum-packed foods, modified-atmosphere-packed food (with reduced oxygen levels), and acidic foods such as salad dressings, processed meat, and fruit or tomato-based products. YM are intentionally added to some foods, such as bread and mould-ripened cheeses. However, they can also contribute to food spoilage, particularly at the end of a product’s shelf life, mainly in relatively dry and acidic products.

We compiled a food microbial composition table for foods present in the Dutch Food Composition table (NEVO) [30] (Appendix A). Foods were grouped into 99 categories based on similar preparation methods and microbial levels (e.g., skimmed milk, semi-skimmed milk, and full-fat milk were combined). For each food group, the minimum, best, and maximum levels of TCB, LAB, and YM (CFU/g food) were estimated by two expert food microbiologists, supplemented with experimental data from Wageningen University (unpublished data), and cross-verified with available literature data [31,32,33]. The minimum estimate represents microbial levels in food at the beginning of its shelf life or immediately after preparation, such as cooking. The best estimate reflects the average microbial levels at the time of consumption, while the maximum estimate corresponds to levels at the end of the food’s shelf life.

### 2.3. Estimation of Food Microbial Intake in the Study Population

To estimate the number of food microbial intake of the study population, the average food intake in grams or ml was calculated based on the three 24-HDRs. This intake was then multiplied by the estimated microbial load of each food, as obtained from the microbial composition table (CFU/g) for TCB, LAB, and YM. These microbial loads were applied for the minimum, best, and maximum estimates. The total microbial intake was calculated by summing the contributions from all foods (Figure 1).

Also, the intra- and interindividual variations in food microbial intake were assessed based on the three-day 24-HDRs of the participants for minimum, best and maximum levels of total TCB, LAB, and YM intakes.

### 2.4. Identification of Major Contributing Foods to Total Microbial Intake and Its Variation

To identify the foods most importantly contributing to the total level and variation of the microbial intake, the average daily intake of each study participant was compared to the best and maximum estimated levels of microbial intake. The average daily intake of each study participant was also compared to the variation in microbial intake across all participants. Contributions to microbial exposure levels and their variation were calculated for the 99 food groups in the database (Appendix A). Furthermore, key foods were separately identified for exposure to TCB, LAB, and YM.

### 2.5. Collection of 24 h Duplicate Foods

From each study participant, duplicate food samples representing 24 h of consumption were collected to determine the microbial content in the lab [29]. These collections were conducted during the same period as the 24-HDRs. Participants collected duplicate portions of all foods and drinks consumed during 24 h using household measures. Each participant received an electric cool box to store the duplicate samples at 5 °C. Participants were instructed, verbally and in print, on how to collect identical second portions of all foods and drinks. Collected samples were retrieved within 24 h after collection, then weighed and homogenised in a commercial blender. The homogenised samples were stored at −20 °C until further quantitative microbial analyses.

### 2.6. Evaluation of Estimated Food Microbial Intake Using Conventional Culturing Techniques (Plate Counting) of Duplicate Foods

The 24 h duplicate food samples from the study population [29] were defrosted and analysed in the laboratory to quantify the microbial content through culturing techniques (Appendix A). The microbial intake estimated from the 24-HDRs was then compared with the microbial plate counts obtained from the duplicate foods.

### 2.7. Statistics

IBM SPSS Statistics 26 was used for statistical analyses. To assess interindividual variation in microbial exposure, a one-way ANOVA was conducted using each participant’s mean microbial intake (calculated by multiplying intake from three 24 HDRs by the estimated microbial load). Separate analyses were performed for minimum, best, and maximum intake estimates.

Intraindividual variation in microbial intake was analysed by two-way ANOVA, considering the effects of the three assessment days (day 1, day 2, and day 3) and the three microbial estimates (minimum, best, and maximum). This analysis compared the mean microbial exposure across individual 24-HDR for minimum, best, and maximum estimations among the 34 participants. This illustrated variation in microbial intake across different days.

To select foods contributing most to the level and variation of the total dietary microbial load, we applied the so-called MOM2 analysis. This procedure is also used to identify foods to generate a food frequency questionnaire [34]. We computed the total explained variance of these foods to the total dietary microbial exposure using regression analyses.

The agreement between the microbial intake estimated from 24-HDR and microbial counts determined in duplicate foods was evaluated using Bland–Altman plots, including limits of agreement.

## 3. Results

### 3.1. Estimated Individual Food Microbial Intake at Minimum, Best, and Maximum Levels

The mean total food microbial intake at the best estimation level was 9.5 log CFU per day (3 billion organisms, i.e. 3.2 × 10^9^), with a range of 7.2–10.4 log CFU/day. Specifically, the mean intake of TCB was 7.3 log CFU/day, ranging from 4.8 to 8.7 log CFU/day. For LAB the mean was 9.2 log CFU/day, ranging from 5.1 to 10.4 log CFU/day. For YM, the mean was 6.0 log CFU/day with a range of 5.0–7.8 log CFU/day, as shown in Table 1. When considering the minimum and maximum levels, the ranges were much larger: 5.7–11.6 log CFU/day for total food microbial exposure, 2.5–11.4 log CFU/day for TCB, 3.4–11.5 log CFU/day for LAB, and 2.6–9.6 log CFU/day for YM (Appendix A). Figure 2A–D illustrate these ranges for total food microbial intake (Figure 2A), TCB (Figure 2B), LAB (Figure 2C), and YM (Figure 2D) for all individual subjects (*n* = 34) per day. These ranges are shown at minimum, best, and maximum levels, as averaged over three 24-HDRs.

### 3.2. Estimated Intraindividual Variation in Food Microbial Intake at Minimum, Best, and Maximum Level

Significant variation in microbial intake was found within each study participant when considering the minimum, best, and maximum microbial estimates. However, no significant difference in microbial exposure was observed between the different days of intake, as indicated by the lowercase alphabetic notations in the data in Appendix A. A substantial difference was noted between the minimum and maximum estimates of microbial exposure.

Figure 3A–D illustrate the intraindividual variation in daily food microbial intake across participants, based on data from three-day 24 h dietary recalls. This intraindividual variation is estimated from the amounts of the various food consumed (in grams) and their minimum, best and maximum microbial estimates. The differences between the minimum, best, and maximum result from infrequently consumed foods at different stages of shelf-life. These figures depict intraindividual variation vertically for all participants, highlighting that the intraindividual variation is more pronounced than the interindividual variation (variation along the *x*-axis), ranging from 4.2 to 11.6 log CFU for TCB (Appendix A).

### 3.3. Determination of Food Microbial Content of 24 h Duplicate Foods Using Plate Counting as a Reference

Microbial levels measured in 24 h (1-day) duplicate food sampling fell within the range of the estimated levels, as depicted in Figure 3A–D (*n* = 34).

Total microbial content in the duplicate foods ranged from 6.4 to 11.0 log CFU/day (Figure 3A). The intake of LAB ranged from <1 to 10.3 log CFU/day, TCB from 5.4 to 11.0 log CFU/day, and YM from <1 to 7.9 log CFU/day.

LAB in duplicate portions generally ranged between the minimum and best estimated levels (Figure 3C), except for one participant whose LAB measurement was below the detection limit. For TCB, measured levels generally ranged between the best and maximum estimated levels (Figure 3C). For some study participants, measured YM levels (Figure 3D) were within the range of best and maximum estimations, while others had YM measurements below the detection limit.

### 3.4. Agreement Between Estimated and Measured Microbial Content

The Bland–Altman plot (Figure 4) displays the average of the best estimates and the measurement in duplicate portions against the difference between the two methods. The Figure 4 shows that, on average, the best estimated total microbial exposure was 0.6 log CFU higher than values determined by plate counting of the duplicate foods. Lower estimates tended to overestimate microbial content, while higher estimates seemed to underestimate it. The limits of agreement ranged from +3 to −2 log (CFU).

### 3.5. Identification of Foods Contributing Most to the Level and Variation of Microbial Intake

The MOM2 analysis, based on the best and maximum estimates of food microbial load, resulted in a list of 28 identified food groups (Table 2) out of 99 food groups in the database (Appendix A), which contributed to over 95% of the variance in total dietary microbial exposure. Of these twenty-eight food groups, certain foods were primary contributors to both the total intake and the variation in intake of specific group of microbes.

For the best estimates of LAB intake, fermented dairy products, specifically yoghurt, cheese, and quark, accounted for the majority intake of LAB. Furthermore, they also explained a significant portion of the variance.

For maximum estimates of LAB intake, in addition to buttermilk, mouldy cheese, and salami/other fermented meats and sliced meats contributed most to the intake level and explained a major part of the variance of intake of LAB. For both the best and maximum estimates of TCB, raw fish, raw vegetables, cold meals, and sliced meats contributed most to the level of total intake and variance.

Lastly, for the best and max estimates of YM, mould-ripened cheese, fresh unpeeled fruit, peeled fruit, fruit salads, and beer/cider contributed most to the level of total intake and variance in exposure.

## 4. Discussion

This study demonstrates significant variation in microbiological exposure, both within and between individuals, with the within-individual variation being the highest. This was established by combining food intake collected by 24 h dietary recalls with estimated microbial contents of foods. These estimations account for various food preparation methods and storage durations. The fact that intraindividual variation is larger than interindividual variation suggests that storage methods (differences between min and max) may play a more significant role than dietary consumption habits. Evaluation by measured values of food microbes in duplicate portions showed a good agreement with estimated intakes by 24 h recalls as nearly all measured values fell within the ranges of estimated food microbial intakes. The total microbial intake in these Dutch participants was primarily determined by LAB.

When taking into account the estimated minimum and maximum levels, the mean interindividual differences in total food microbial intake ranged from 5.7 to 11.6 log CFU/day. This corresponds to approximately half a million (5.4 × 10^5^) to more than 400 billion (4.4 × 10^11^) log CFU/day, representing a difference of about a factor of 1 million in exposure to food microbes per day. This wide range was confirmed by plate counting of the microbial content in duplicate foods, which ranged from 6.4 to 11.0 (2.8 × 10^6^ to 9.4 × 10^10^) log CFU/day (3 million to 100 billion), or a difference of about 5.0 log CFU/day (a factor of 100,000) and closely aligned with the estimated lower and higher levels.

The agreement between the estimated microbial intake from 24 h dietary recalls and the determined plate counting of duplicate food samples suggests quite a good agreement at the population level. Figure 4 shows that the best estimated total microbial exposure was, on average, 0.6 log CFU higher than the value determined by plate counting in the duplicate foods. However, the limits of agreement, representing individual differences, were quite large. In addition, lower estimates tended to overestimate microbial content, while higher estimates seemed to underestimate it [35]. It is also important to note that we did not determine the microbial load of the 24 h duplicate foods on the same day as the collection of the 24-HDRs. Nevertheless, all data were collected within the same study period, representing a similar food pattern to that captured in the 24-HDRs.

Twenty-four food groups accounted for 95% of the variance in total food microbial content in our study population. The most important foods included fermented dairy products, such as yoghurt, buttermilk, cheese and quark, mould-ripened cheese, raw vegetables, raw fish, peeled and unpeeled fresh fruit, fermented and sliced meats, (raw) milk, and beer/cider. These key foods may vary per country, depending on dietary patterns and food habits.

Our data showed that microbial intake was strongly influenced by the types of foods consumed. In addition, methods of food preparation and storage contributed to variation in microbial intake, as indicated by the estimated minimum, best, and maximum intake levels. Interestingly, several foods yielded a significant contribution to microbial exposure even when consumed in small portions. For example, just 1 g of yoghurt provides an exposure of approximately 8 log (±10^8^) CFU bacteria, primarily LAB. In contrast, heat-treated foods, such as canned foods and freshly baked bread, contain low numbers of microbes. Consuming 100 g of heat-treated foods provides only 5 log (±10^5^) CFU, a factor of 1000 lower than the amount in 1 g of yoghurt, making them much less relevant in terms of total microbial exposure. In Table 3, sample menus with minimum and maximum microbial content are given. It shows that a healthy diet consisting of fresh fruits, raw vegetables, yoghurt, and cheese is also rich in food microbes, specifically LAB. However, it should be noted that consumption of more raw foods and those near their expiration date increases the risk of foodborne infections or intoxications. Specifically fresh foods, such as pre-cut vegetables, fresh meat and meat products, fresh or smoked fish, and freshly prepared ready meals can become microbially spoiled if stored beyond their shelf life or under improper conditions (e.g., excessive warmth). This can lead to serious health implications and impact food safety. These foods should not be consumed beyond the expiration date, and special attention should be paid when they are used for vulnerable consumers, such as infants, sick people, and elderly people.

This study is one of the first to quantitatively analyse microbial intake in individuals. A similar study in the USA [21] quantitatively determined the microbial content of 15 meals representing three different typical dietary patterns. Based on plate counts, the USDA meal plan that recommended for emphasising fruits and vegetables, lean meat, dairy, and whole grains had the highest total amount of microbes at 9.1 log (1.3 × 10^9^) CFU per day. This was followed by the VEGAN meal plan at 6.7 log (6 × 10^6^) CFU/day and the AMERICAN meal plan, characterised by convenience foods, at 6.1 log (1.4 × 10^6^) CFU per day. In a second study using cross-sectional data from NHANES (2001–2018), the intake of foods with low (10^4^ CFU/g), medium (10^4^–10^7^ CFU/g), or high (>10^7^ CFU/g) levels of microbes per gram food was assessed in children/adolescents and adults [28]. On average, children and adolescents consumed ∼85 g/day of foods with medium microbial levels and 127 g/d with high levels, with intake slightly increasing over the years. Both the American studies and our research show that a healthy diet contains more food microbes than a diet rich in highly processed foods. The American data from Lang [21] align closely with the microbial levels we assessed from 24-HDR and measured in the duplicate portions in our study.

Food preparation and dietary habits in the Western world have changed significantly in recent decades. On one hand, there has been a shift in homemade meals to convenience meals and processed foods [36], which often undergo processing or storage methods that reduce or eliminate food microbes. Conversely, there has been an increase in the consumption of raw and fermented products [18,20,36]. Consequently, these modified food habits have likely caused shifts and significant variations in the quantity of food microbes consumed. Such changes may have an impact on the intestinal microbiome and its homeostasis, potentially impacting the development or treatment of allergic disease.

Based on current knowledge of live food microbes, the health implications of microbial intake for allergic disease are promising but still need further investigation. In a scoping review, Iyer et al. [19] found that the intake of ≥2 × 10^9^ food microbes was associated with positive health effects in various conditions, including respiratory health.

A strength of our study is the creation of a food microbial table (Appendix A) for estimating food microbial intake. The estimated minimum, best, and maximum values provided by the food microbiologists were based on established microbial knowledge and literature [31,32,33], supplemented with lab measurements. However, further work is needed to validate these estimations of food microbial intake, explore food microbial differences between dietary patterns [21], compare with other databases [28], assess microbial content changes throughout shelf life in household settings, and examine the influence of storing conditions. When validated, the food microbial load table is expected to be a valuable instrument for future studies, including large-scale epidemiological studies, to evaluate associations between microbial exposure and various health outcomes.

## 5. Conclusions

In conclusion, the significant intra- and interindividual variation in the intake of live food microbes among Dutch adults we found in this study may be of clinical relevance.

This high variation in intake was attributed to differences in food choices, preparation methods, and storage conditions. Individuals consuming fermented (dairy) products, cheese, fresh foods, and foods closer to the end of their shelf life had higher intakes of food microbes, while individuals using pre-packed foods and highly processed foods had lower intakes of food microbes. These differences may significantly influence the gut microbiota composition, inflammation, and consequently the prevention or treatment of allergic diseases. A low intake of food microbes may be a risk factor for prevention and treatment of allergic diseases, while a high intake may contribute to prevention and treatment.

## Figures and Tables

**Figure 1 nutrients-17-01248-f001:**
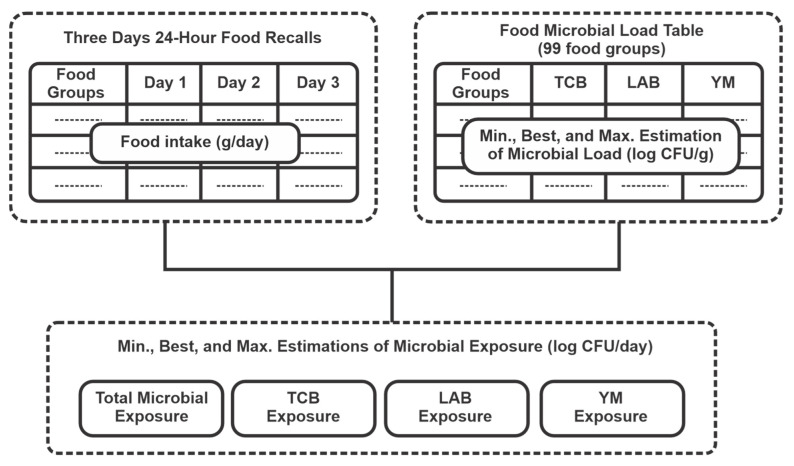
Quantitative estimation of food microbial content using 24 h dietary recalls and a food microbial content table.

**Figure 2 nutrients-17-01248-f002:**
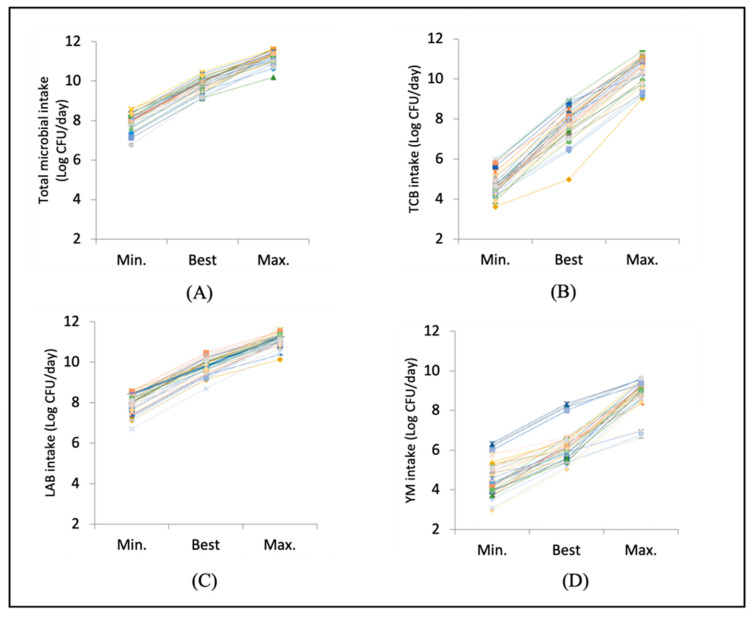
Ranges in interindividual total food microbial intake (**A**), TCB (**B**), LAB (**C**), and YM (**D**) per day, as estimated from the average values of three 24 h HDRs (*n* = 34, all 34 individuals shown in different line/colour/symbol) and presented for minimum, best, and maximum intakes.

**Figure 3 nutrients-17-01248-f003:**
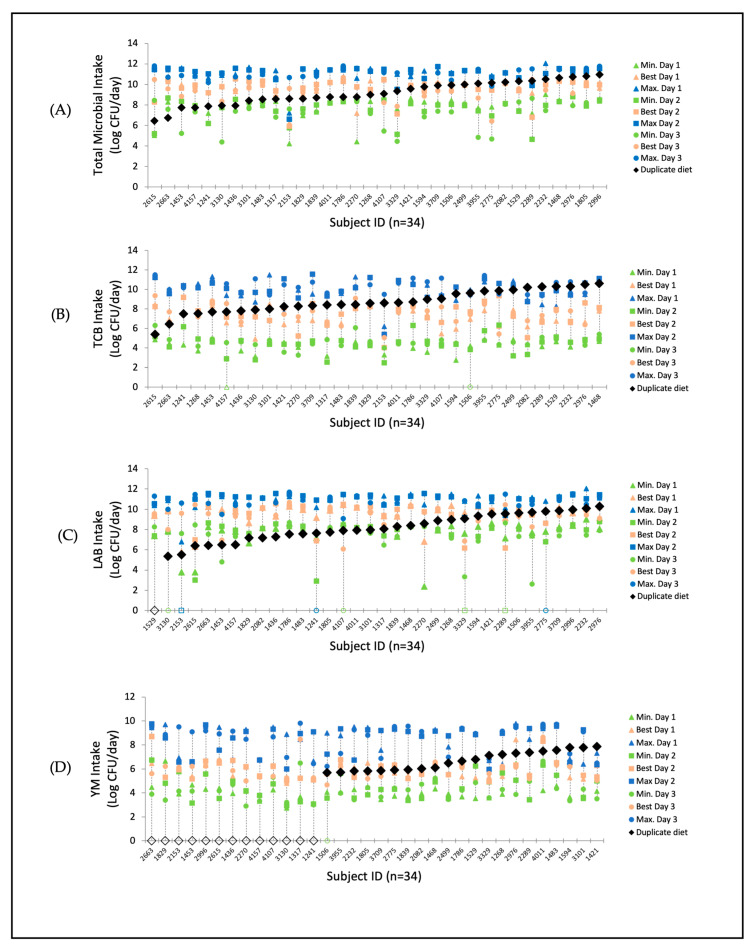
Estimated intake from 24 h dietary recalls of each study participant (*n* = 34) of total food microbes (**A**), TCB (**B**), LAB (**C**), and YM (**D**) and the measured intake of 1-day duplicate portion per person. The intraindividual range of estimated intake of days 1, 2, and 3 is presented with minimum levels (green), best levels (pink), and maximum levels (blue). The measured intake from 1-day duplicate portions is presented by black diamonds (open diamonds represent results below the detection limit around 4 log CFU/g).

**Figure 4 nutrients-17-01248-f004:**
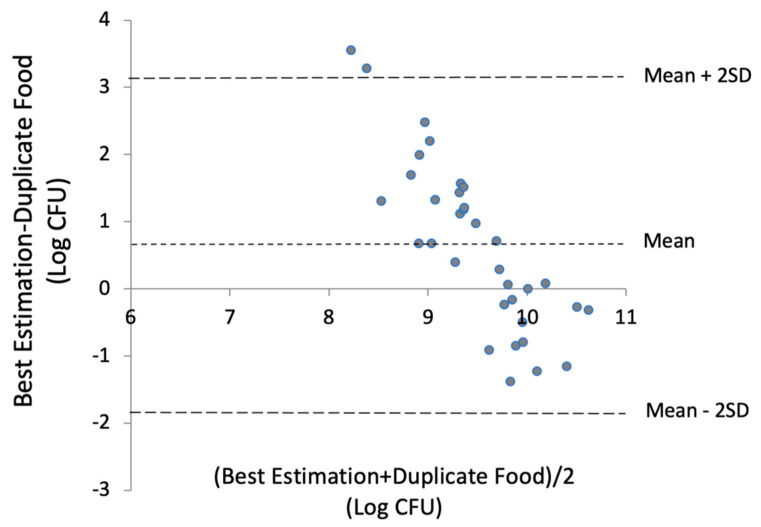
Bland–Altman plot of the mean difference and limits of agreement between the best estimations from 3-day 24 h recalls and the microbial content of one 24 h duplicate portion compared to the average of the two methods of the 34 study participants.

**Table 1 nutrients-17-01248-t001:** Estimated individual daily microbial intake based on three days-24 hRs (**A**) and measured intake of one day duplicate portion per person (**B**).

A. Mean of Estimated Individual Food Microbial Intake Based on Three 24 hRs (*n* = 34)				
Group of Microbes	Day	Mean and Range of Microbial Exposure (Log CFU/Day)		
		Minimum Estimation	Best Estimation	Maximum Estimation
Total Contaminating Bacteria (TCB)	1	4.2 (0–4.9) ^Aa^	7.0 (4.9–8.5) ^Ba^	9.8 (6.2–11.5) ^Ca^
	2	4.5 (2.5–6.4) ^Aab^	7.3 (0.3–9.4) ^Bab^	10.1 (5.4–11.6) ^Cab^
	3	4.6 (0–6.3) ^Ab^	7.6 (5.1–9.4) ^Bb^	10.4 (9.3–11.5) ^Cb^
	Average	4.4 (2.5–5.6) *	7.3 (4.8–8.7) *	10.1 (7.0–11.4) *
Lactic acid bacteria (LAB)	1	7.4 (0–9.0) ^Aa^	9.4 (5.8–10.5) ^Ba^	10.9 (6.8–12.0) ^Ca^
	2	7.3 (0–8.8) ^Aa^	9.3 (0–10.5) ^Ba^	10.7 (0–11.6) ^Ca^
	3	6.5 (0–9.1) ^Aa^	8.7 (0–10.7) ^Ba^	10.1 (0–11.7) ^Ca^
	Average	7.0 (3.4–8.5)	9.2 (5.1–10.4) *	10.6 (5.8–11.5)
Yeasts and Mould (YM)	1	4.1 (2.7–6.7) ^Aa^	6.0 (4.8–8.7) ^Ba^	8.1 (4.4–9.8) ^Ca^
	2	4.5 (2.9–6.7) ^Aa^	6.0 (5.0–8.7) ^Ba^	8.3 (3.9–9.7) ^Ca^
	3	4.1 (0–9.4) ^Aa^	6.0 (4.7–8.7) ^Ba^	8.3 (4.5–9.8) ^Ca^
	Average	4.2 (2.6–5.7)	6.0 (5.0–7.8) *	8.4 (6.5–9.6) *
Total microbial exposure	1	7.7 (4.2–9.0) ^Aa^	9.6 (6.2–10.5) ^Ba^	11.1 (7.2–12.0) ^Ca^
	2	7.6 (4.6–8.8) ^Aa^	9.6 (5.9–10.5) ^Ba^	11.0 (6.6–11.7) ^Ca^
	3	7.2 (4.4–8.7) ^Aa^	9.4 (6.4–10.7) ^Ba^	11.1 (9.8–11.8) ^Ca^
	Average	7.5 (5.7–8.5)	9.5 (7.2–10.4) *	11.1 (8.2–11.6) *
**B. Mean of Measured Food Microbial Intake Based on One Day Duplicate Portion per Person (*n* = 34)**				
**Group of Microbes**		**Mean, Median, and Range of Microbial Exposure (Log CFU/day)**		
Total aerobic bacteria (TAB) **		10.0; 8.6; 5.4–11.0		
Lactic acid bacteria (LAB)		9.4; 8.1; Appr. 4–10.3 ***		
Yeasts and moulds (YM)		7.0; 5.6; Appr. 4–7.9 ***		
Total food microbial intake		10.1; 9.0; 6.4–11.0		

Notes: Three-day 24 hRs: three-day twenty-four hour recalls; CFU/day = colony forming units per day. * Based on the one-way ANOVA test, there are significant interindividual differences among the mean of 34 individuals’ daily microbial exposures based on three-day 24 hRs (*p* < 0.05). Based on Two way-ANOVA with repeated measures, values with different alphabetic notations show statistically significant intraindividual mean differences (*p* < 0.05). Uppercase notations (capital letters A, B, C) indicate the differences between microbial intake level obtained from different estimations (minimum, best, and maximum), while lowercase notations (a, b) indicate the differences between microbial intake data from three different days (day 1, day 2, and day 3) (*p* < 0.05). ** Total aerobic bacteria (TAB) also include lactic acid bacteria (LAB) as well as yeasts and moulds (YM) that can grow in the agar media. *** The lower range of LAB and YM was approximately 4 log CFU/day or lower because the plate counting was only sensitive enough to detect ~4 log CFU per total duplicate sample depending on the sample weight.

**Table 2 nutrients-17-01248-t002:** Foods contributing most to the level and variation of lactic acid bacteria (LAB), total contaminating bacteria (TCB), and yeast/mould (YM) intake in the study population *.

Food Items	Contribution to Exposure Level (%)		Contribution to Variance (%)	
	Best	Max.	Best	Max.
**Intake of LAB**				
Yoghurt	43.4	25.6	65.8	24.8
Cheese (excl. mould cheese)	35.5	21.0	15.0	5.6
Quark, fresh cheese	14.0	8.3	17.1	6.4
Buttermilk	3.6	21.1	1.5	55.4
Drink yoghurt	2.5	1.5	0.6	0.2
Salami and other fermented meats	0.6	3.7	0.0	1.2
Mould cheese	0.3	1.4	0.0	0.3
Meat products (Sliced meats)	0.0	16.2	0.0	6.0
Milk	0.0	0.8	0.0	0.0
**Intake of TCB**				
Raw fish	51.5	1.6	89.8	0.2
Raw vegetables	27.3	83.9	4.0	97.8
Cold meals (uncooked)	14.5	4.4	5.8	1.4
Meat products (Sliced meats)	1.4	4.4	0.0	0.2
Raw milk	1.2	0.4	0.2	0.1
Milk	0.7	2.2	0.0	0.1
Vegetarian meat on bread	0.3	0.1	0.0	0.0
Cut vegetables	0.2	0.5	0.0	0.1
Unpeeled fruits (fresh)	0.0	1.1	0.0	0.0
Peeled fruits (fresh)	0.0	1.2	0.0	0.0
**Intake of YM**				
Mould cheese	93.3	13.5	100.0	28.7
Unpeeled fruits	2.6	37.6	0.0	31.1
Fruits juice	2.5	0.4	0.0	0.0
Bread and bread substitutes (excl. sourdough)	0.6	0.1	0.0	0.0
Peeled fruits	0.3	40.6	0.0	35.9
Savory bread salad	0.2	0	0.0	0.0
Fruit salad (excl. canned fruits)	0.1	1.2	0.0	1.2
Beer, cider	0.0	6.4	0.0	2.9
Margarines	0.0	0.1	0.0	0.0

* Note that data are presented in linear scale and not in log scale.

**Table 3 nutrients-17-01248-t003:** Sample menus with minimum and maximum microbial content are given.

Sample Menu with Low Microbial Content	Sample Menu with High Microbial Content
*Breakfast:*	*Breakfast:*
UHT milk, breakfast cereals	Fresh milk, muesli,
Crackers	fresh fruit
Cheese spread, jam	
*Lunch:*	*Lunch:*
Toasted or bread microwaved for defrosting	Two-day old fresh bread
Canned fish	Raw fish, salami
Canned meat	Fresh salad, sprouts
Ultra-processed foods, e.g., pizza/hot dog	Cheese, mould cheese
*Snacks:*	*Snacks:*
Chips, confectionery	Fresh fruits, fresh olives,
Soda	Fresh orange juice
Chocolate bar	Nuts
*Dinner:*	*Dinner:*
Instant soup	Home cooked soup
Ready meals/meat/sauces	Home cooked dinner
Canned or jarred vegetables/apple sauce	Fermented vegetables, mushrooms, sauerkraut, home-cooked apple sauce
No use of leftovers	Leftovers
Ready-to-use pudding	Yoghurt
*General:*	*General:*
Ultra-processed foods	Home-made foods
Packages not opened until consumption and consumed long before the end of shelf life	Opened packages until the end of shelf life in the fridge

## Data Availability

The raw data supporting the conclusions of this article will be made available by the authors on request.

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
