# Peer review of "Considerable Variation in Intake of Live Food Microbes in Dutch Adults"

_nutrients, 2025, doi:10.3390/nu17071248_

Round 1
Reviewer 1 Report
Comments and Suggestions for Authors
CONSIDERABLE VARIATION IN INTAKE OF LIVE FOOD MICROBES IN DUTCH ADULTS
The study aimed at quantitatively assess the intake of live food microbes in Dutch adults and compare these estimates with those obtained from duplicate portions. Food-microbial content of 34 Dutch adults was assessed for three dominant groups: total contaminating bacteria (TCB), lactic acid bacteria (LAB), and yeasts/molds (YM). The estimated total microbial exposure varied considerably across individuals, ranging from 5.7-11.6 log (5.4x105-4.4x1011) CFU/day. The exposure to TCB ranged from 2.5-11.4 log (3.0x102-2.5x1011) CFU/day, LAB from 3.4- 27
11.5 log (2.3x103-3.0x1011), and YM from 2.6-9.6 log (3.6x102-4.3x109) CFU/day. The authors concluded that the intake of live food microbes among Dutch adults varied considerably, ranging from nearly a million to more than 100 billion per day. Further validation of the food-microbial database is required.
Revise the abstract, correct the numbers for example 3.6x102. Provide a little information about the statistical design, add P value.
L38: Provide more information about allergic diseases.
Try to shorten the introduction, it's too long.
Materials and methods section is well presented
Fig. 3: not clear
Provide a better conclusion of your work based on your findings
Institutional Review Board Statement: provide number if applicable.
Reviewer 2 Report
Comments and Suggestions for Authors
The manuscript presents valuable findings on dietary microbial intake. However, some sentences are too complex and could be restructured for better readability.
The figures are of poor quality, some are unreadable. Please replace them to ensure greater clarity.
The abstract is well structured, well-designed.
The introduction is relevant, and complex.
Section 3.3: Can you clarify how many participants were affected?
Section 3.4: Agreement between estimated and measured microbial content
“Lower estimates tended to overestimate microbial content, while higher estimates seemed to underestimate it.” Why is there this discrepancy? Any possible biases in the estimation?
Section 3.5: Identifying foods that contribute most to microbial intake
Has any sensitivity analysis been performed to confirm the robustness of this selection?
In Table 1 you should add below the table each abbreviation used in the table.
Please correct Table 2 so that each digit is 1 decimal place (instead of 0 being 0.0)
Discussion
"This study demonstrates a significant variation in microbiological exposure, both within and between individuals, with the interindividual variation being the highest..." comes from dietary habits, storage methods or external contamination? Please discuss this in more detail from this point of view.
"The agreement between the estimated microbial intake from 24-hour dietary recalls and the determined plate counts of duplicate food samples suggests a fairly good agreement at the population level". Consider specifying the statistical method used to assess the agreement.
"A healthy diet consisting of fresh fruits, raw vegetables, yogurt and cheese is also rich in dietary microbes, especially LAB". While true, this could be expanded to mention potential health implications or risks.
The conclusion is somewhat speculative. If possible, provide a conclusion that follows from the results.
Round 2
Reviewer 2 Report
Comments and Suggestions for Authors
Dear Authors
The manuscript has been improved, point by point. In this form I recommend it for publication.
Congratulations!